# Electric Current Generation by Increasing Sucrose in Papaya Waste in Microbial Fuel Cells

**DOI:** 10.3390/molecules27165198

**Published:** 2022-08-15

**Authors:** Segundo Rojas-Flores, Magaly De La Cruz-Noriega, Santiago M. Benites, Daniel Delfín-Narciso, Angelats-Silva Luis, Felix Díaz, Cabanillas-Chirinos Luis, Gallozzo Cardenas Moises

**Affiliations:** 1Vicerrectorado de Investigación, Universidad Autónoma del Perú, Lima 15842, Peru; 2Grupo de Investigación en Ciencias Aplicadas y Nuevas Tecnologías, Universidad Privada del Norte, Trujillo 13007, Peru; 3Laboratorio de Investigación Multidisciplinario, Universidad Privada Antenor Orrego (UPAO), Trujillo 13008, Peru; 4Escuela Académica Profesional de Medicina Humana, Universidad Norbert Wiener, Lima 15046, Peru; 5Instituto de Investigación en Ciencias y Tecnología de la Universidad Cesar Vallejo, Trujillo 13001, Peru; 6Universidad Tecnológica del Perú, Trujillo 13011, Peru

**Keywords:** saccharose, microbial fuel cells, waste, papaya, bioelectricity

## Abstract

The accelerated increase in energy consumption by human activity has generated an increase in the search for new energies that do not pollute the environment, due to this, microbial fuel cells are shown as a promising technology. The objective of this research was to observe the influence on the generation of bioelectricity of sucrose, with different percentages (0%, 5%, 10% and 20%), in papaya waste using microbial fuel cells (MFCs). It was possible to generate voltage and current peaks of 0.955 V and 5.079 mA for the cell with 20% sucrose, which operated at an optimal pH of 4.98 on day fifteen. In the same way, the internal resistance values of all the cells were influenced by the increase in sucrose, showing that the cell without sucrose was 0.1952 ± 0.00214 KΩ and with 20% it was 0.044306 ± 0.0014 KΩ. The maximum power density was 583.09 mW/cm^2^ at a current density of 407.13 A/cm^2^ and with a peak voltage of 910.94 mV, while phenolic compounds are the ones with the greatest presence in the FTIR (Fourier transform infrared spectroscopy) absorbance spectrum. We were able to molecularly identify the species Achromobacter xylosoxidans (99.32%), Acinetobacter bereziniae (99.93%) and Stenotrophomonas maltophilia (100%) present in the anode electrode of the MFCs. This research gives a novel use for sucrose to increase the energy values in a microbial fuel cell, improving the existing ones and generating a novel way of generating electricity that is friendly to the environment.

## 1. Introduction

Due to the exponential increase in society, it has generated two main problems, the need for new sources of electricity generation and a way to reuse the waste produced by human consumption. This last problem has generated a problem for the collection centers of the different municipalities of large and small cities [1,2]. In 2016, waste production exceeded 2.02 billion tons, and it was estimated that by 2050 it would be approximately 3.4 billion tons, which will make waste management increase from 1.61 million dollars in 2020 to 2.50 million by 2030 [3,4]. In this sense, part of the waste is that from agricultural production, which in its process of sowing, harvesting, sale and consumption, generates different types of waste that in recent years have begun to be reused in order to give them a second use and take advantage of them in other activities of society [5]. One of the waste products with the highest production in Latin America and consumed worldwide is papaya derivatives (*Carica papaya* L.). In its different presentations, this product represents approximately 15.36% (11.22 metric tons) of tropical fruits produced each year; approximately 24 countries produce this type of product [6]. The high consumption of this fruit is mainly due to the high amounts of vitamins A, C and E that it presents, besides being a natural diuretic. In the last decade, its production has increased by 85% for South American countries, mainly in their tropical zones [7]. On the other hand, new technologies have emerged to generate electricity in a sustainable way, for example, microbial fuel cells (MFCs). These types of devices generate electricity through the oxidation and reduction processes that occur inside them, converting chemical energy into electrical energy [8,9]. These systems are generally composed of two chambers (anodic and cathodic). In the chamber where the anodic electrode is placed, the microorganisms oxidize the organic matter, producing electrons, which travel through an external circuit to the cathodic chamber where they are oxidized. Due to this process, a flow of electrons occurs, producing electricity [10,11,12].

There is a wide variety of substrates used as fuels in MFCs, with biological factors being a fundamental characteristic because they are used as an environment for microbial growth and other metabolic activities, therefore, the composition and degradability of the substrate promote the rate of activity for the generation of electrons, which translates into a better performance of the MFCs [13,14,15]. Various investigations have reported the use of a wide range of substrates, from domestic, industrial and municipal wastewater to simpler substrates such as glucose, which serves as a carbon source in MFCs for electricity production [16,17]. In this sense, the concentration of glucose as a substrate establishes the maximum amount of chemical energy available to convert into electrical energy, so a substrate with a high content of sugars can improve the generation of electrons in MFCs [18]. In this sense, in their research, Kamau et al. (2020) used waste from avocados, tomatoes, bananas, watermelons and mangos as substrates, mainly monitoring the voltage and current values generated in MFCs, reporting that the tomato produced the highest voltage (0.702 V) and, in terms of current values, it increased linearly over time for all residues. On the other hand, that study indicated that moisture content and carbohydrate level were the main factors influencing electricity generation [19]. Likewise, Kalagbor and Akpotayire (2020) evaluated the generation of electricity from tropical fruit residues (watermelon and papaya) in single-chamber MFCs. The cells were monitored for a period of four weeks, and the maximum voltage generated was 139.5 mV in the use of the watermelon substrate and 222.9 mV produced by the papaya. In terms of the power density of the watermelon substrate, it was 0.2452 mW/cm^2^. For the papaya substrate, on the other hand, the values of dissolved oxygen (DO) and biological oxygen demand (BOD) showed that the medium was conducive to the proliferation of microorganisms [20,21]. These results demonstrated that single-chamber MFCs are capable of generating electricity from tropical fruit residues, so the use of these systems was recommended as a sustainable alternative since they represent an option to increase electricity supply in urban and rural areas [22,23]. Likewise, Utami and Yenti (2018) studied the generation of electrical energy from papaya peel waste in MFCs. In the anaerobic compartment of the anode, the layer of this substrate was used as an electron donor, while in the cathode compartment, KMnO_4_ was used as an electron acceptor. Regarding the results, the power density was 121 mW/m^2^ and a current of 179 mA with a voltage of 1.095 V [24]. In this sense, it has been proven that carbohydrates rich in glucose and fructose are of essential importance for the generation of electricity in MFCs. In this sense, sucrose is a natural component present in all-natural juices.

In this sense, the main objective of this research was to evaluate the generation of electricity using papaya residues as a substrate by adding different concentrations of sucrose (0%, 5%, 10%, and 20%) in a single-chamber microbial fuel cell manufactured at low cost with Zn-Cu electrodes, monitoring their voltage, current, current density, power density and pH for 30 days. Thus, the values of the internal resistance and absorbance spectrum were also measured by FTIR (Fourier transform infrared spectroscopy).

## 2. Results and Discussion

Figure 1 shows the influence of sucrose concentrations on the generation of voltage, current and pH, with the concentration of sucrose at 20% being the one with the highest value when used as a substrate, reaching a maximum value (0.955 V) on day 15. From the first day, the measurements were remarkable, with the highest voltage values when the sucrose concentration was 20% generating 0.19 V more than the MFCs with 10% and 0.26 V with the MFCs used as blank (Figure 1a). While in Figure 1b it is observed that the MFCs with 20% sucrose generated a higher electric current with a peak value of 5.079 mA on the sixteenth day, all electrical current values have their maximum peaks between the eleventh and sixteenth day. The main reason for the increase in current parameters, according to Fujimura et al. (2022), is because sucrose, being a disaccharide (glucose and fructose), has been used for fermentation, releasing electrons in the process, generating electric current [24]. On the other hand, when glucose is coupled in respiratory chains, it is oxidized to gluconate by glucose dehydrogenase and is subsequently oxidized to 2-cetagluconate by gluconate dehydrogenase [25]. While microorganisms consume glucose as a source of carbon electrons and protons, previous research mentions that 24 mol of electrons and hydrogen ions are generated by the oxidation of one mole of glucose under anaerobic conditions [26]. On the other hand, the pH values increased from the first day of monitoring to the last, as shown in Figure 1c; the optimum operating pH of 4.45, 4.61, 4.77 and 4.98 for the MFCs with 0%, 5%, 10% and 20% sucrose on days 17, 15, 14 and 15, respectively. Leiva et al. (2018) mention that the low pH values in microbial fuel cells are due to the accumulation of protons in the anode electrode and hydroxide ions in the cathode electrode [27]. The pH values influence the generation of voltage and current in MFCs mainly because the microorganisms present in each cell need the ideal conditions for their growth and acclimatization [28].

Figure 2 shows the values of the electrical resistance obtained from the microbial fuel cells at different percentages of sucrose, where the experimental data is adjusted to Ohm’s law (V = RI), where the *x*-axis is the current (I) and the y axis is the voltage (V), for which the slope of the linear fit is the internal resistance (R_int._) of the cells. The R_int._ values found were 0.044306 ± 0.0014, 0.03572 ± 0.00716, 0.02269 ± 0.0015 and 0.1952 ± 0.00214 KΩ and stop the MFCs with 0%, 5%, 10% and 20% sucrose. As clearly observed in Figure 3, the values decrease with increasing sucrose concentration. According to Ueda et al. (2022), the time required for the decomposition of the substrates has a dependence on the resistance of the microbial fuel cells. It is known that when the resistance is low, the electrons flow more freely, generating a greater electric current, and it is probable that this affects the microbes in the electrode biofilm [29,30].

Figure 3 shows the values of power density (PD) and maximum voltage as a function of current density (CD), being the MFCs with 20% sucrose the one that generated a higher value of PD with 583.09 mW/cm^2^ at a CD of 407.13 A/cm^2^ and a peak voltage of 910.94 mV; with a 36% higher than that generated by the MFCs used as target (0% sucrose) that generated a PD of 427.14 mW/cm^2^ in a CD of 4.920 A/cm^2^ with a peak voltage of 537.72 mV. These values are higher than those generated by Mohamed et al. (2020), where they used kitchen wastewater and photosynthetic algae as fuel in their dual-chamber MFCs, managing to generate maximum PD peaks of 31.6 ± 0.5 mW/cm^2^ in a CD of 172 mA/cm^2^ with a maximum voltage of 600 mV [31]. In the same way, Kondaveeti et al. (2019) used citrus peels as fuel in their single-chamber MFCs, managing to generate PD peaks of 63.4 mW/cm^2^ at a CD of 280.56 mA/cm^2^ at a peak voltage of 0.478 V [32]. According to Yaqoob et al. (2020) obtained high values of PD shown in the research are due to the metallic electrodes used due to the good electrical properties they have, so the current losses are few in the energy generation process [33].

Figure 4 shows the absorbance spectrum of the compounds present in the different substrates (0%, 5%, 10% and 20% sucrose), observing the most intense peak at 3289 cm^−1^ belonging to the N-H stretch, O-H groups, phenols, and carboxylate acids, while peaks 2904 and 2848 cm^−1^ are associated with the C-H stretching of alkanes, aldehydes and ketones. In the same way, the peak of 1756 cm^−1^ belongs to alkane C-H stretching, while 1660 is associated with C=C stretching, N-H primary amine, C=N stretching and amide stretch. The 1545 cm^−1^ peak indicates the presence of alkane C-H stretching, alkene C=C stretching, C=N stretching, primary and secondary amine C-N stretching and amide; and finally, the peaks at 1255 and 1030 cm^−1^ alkane C-H stretching, alkene C=C stretching, C=N stretching, primary and secondary amine C-N stretching and amide [34,35,36]. It has been shown that the high content of phenols releases large amounts of electrons which travel through the external circuit to the cathode electrode, thus generating a higher electrical current output [37,38].

Table 1 shows the regions sequenced and analyzed in the BLAST program in which an identity percentage of 99.32% was obtained, which corresponds to the Achromobacter xylosoxidans species, 99.93% to the Acinetobacter bereziniae species, and with 100.0% to the species Stenotrophomonas maltophilia. Figure 5 shows the dedongram, which was built using the MEGA program, which relates and groups sequences of species [39]. These bacteria are ubiquitous, they are found in soil, water, air, plants and animals. They transfer electrons to the anode via external loop carrier proteins, such as cytochrome c, or via membrane appendages called [40,41] nanowires. An essential factor in the production of electric current is the formation of biofilms on the anode electrode. This consists of two types of microorganisms, fermentative and electrogenic. Where the former hydrolyzes organic compounds and the metabolites they secrete and are used as substrates for electrogenic bacteria to generate electrons, protons and CO_2_ through oxidative processes [42]. Figure 6 shows the electricity generation process through microbial fuel cells, where MFCs with 5, 10 and 20% sucrose connected in series were used; managing to generate a voltage of 2.09 V, enough to turn on a red LED bulb. This shows that papaya residues have great values for the generation of bioelectricity. Recent research has shown the importance of other residues in other changes, which leads to the sustainability of these types of products [43,44].

## 3. Materials and Methods

### 3.1. Fabrication of Single-Chamber Microbial Fuel Cells

For the chambers of the microbial fuel cells (three in total), 400 cm^3^ polyethylene terephthalate cubic containers were used, to which an 18 cm^2^ hole was made on one of the faces in which the electrode was placed, cathodic (Zinc, Zn), while the anodic electrode (Copper, Cu) was placed inside the container; both electrodes were joined by means of an external circuit with a resistance of 100 Ω. As a proton exchange membrane, 10 mL of the solution obtained from 6 g of KCl and 14 g of agar in 400 mL of H_2_O were used (see Figure 7). While the preparation of sucrose was carried out at 0 (target), 5, 10, and 20%, for this a 50% sucrose stock solution and papaya residue extract were used, with the final working volume being 200 mL.

### 3.2. Collection of Papaya Waste

Three decomposing papayas (approximately 5 kg) were collected from La Hermelinda market, Trujillo, Peru. Which were collected in hermetic bags and transferred to the laboratory for use where they were washed three times with distilled water to remove any type of impurities (sand, dust or insects). These wastes were ground in an extractor (Labtron, LDO-B10-USA) until obtaining homogeneity throughout the substrate and then stored in a 1000 mL bottle at 20 ± 2 °C until used in microbial fuel cells.

### 3.3. Characterization of Microbial Fuel Cells

The electrical parameters of current, voltage, power density and current density were measured using a multimeter (Prasek Premium PR-85, Chicago, IL, USA) using the method described by Rojas-Flores et al. (2021), whose external resistances were 10 ± 0.2, 40 ± 2.3, 50 ± 2.7, 100 ± 3.2, 300 ± 6.2, 390 ± 7.2, 560 ± 10, 680 ± 12.3, 820 ± 14.5, 1000 ± 20.5 Ω [45]. While the internal resistance was found using the energy sensor (Vernier- ±30 V & ±1000 mA, USA). Likewise, the values of pH and electrical conductivity were monitored with a pH-meter (110 Series Oakton, Chicago, IL, USA) and a conductivity meter (CD-4301, Chicago, IL, USA) during the 30 days of operation. The initial and final transmittance values were measured by FTIR (Thermo Scientific IS50, Chicago, IL, USA).

### 3.4. Molecular Identification of Microorganisms by Sequencing the 16S rRNA Genes

Molecular identification was carried out by the Analysis and Research Center of the “Biodes Laboratories”. From pure or axenic cultures of bacteria, which were based on DNA extraction using the CTAB extraction method, which were analyzed molecularly by amplification of the 16S rRNA gene [46]. The genetic sequences were evaluated with the bioinformatic program MEGA-X to generate consensus sequences and develop phylogenetic trees. The identification of the microbial species was carried out using the Gen-Bank databases and the programs Nucleotide Blast (Basic Local Alignment Search Tool) and EzBio-Cloud [47,48]. The molecular analysis was analyzed only from the MFC with papaya waste with 20% sucrose.

## 4. Conclusions

Bioelectricity was successfully generated using papaya waste with sucrose in different percentages (0%, 5%, 10%, and 20%) as fuel through laboratory-scale microbial fuel cells using zinc and copper as electrodes. The cell that obtained the best electrical parameters was the one that contained the highest percentage of sucrose (20%), managing to generate an electrical voltage and current of 0.955 V and 5.079 mA, respectively, with an optimal operating pH of 4.98 on the fifteenth day. Likewise, the internal resistance of the cells decreased as sucrose increased, with the maximum internal resistance being 0.044306 ± 0.0014 KΩ and the minimum being 0.1952 ± 0.00214 KΩ belonging to the cells with 0 and 20% sucrose, respectively. Thus, it was also observed that the maximum power density was 583.09 mW/cm^2^ at a current density of 407.13 A/cm^2^ with a peak voltage of 910.94 mV, belonging to the cell with 20% sucrose. Finally, the absorbance peaks demonstrate the presence of phenols, which gives indications of the high values of current and voltage. Being able to identify 99.32, 99.93 and 100% of the species Achromobacter xylosoxidans, Acinetobacter bereziniae and Stenotrophomonas maltophilia, respectively, from the anode electrode of the MFCs with 20% sucrose. For future work, replicas (at least three) of each MFC should be made and, using the optimal pH values (4.98) found in this research, standardize the pH, as well as cover the metal electrodes with some chemical compound that is not harmful for the species of microorganisms found (Achromobacter xylosoxidans, Acinetobacter bereziniae and Stenotrophomonas maltophilia species) on the substrates to improve the efficiency of microbial fuel cells.

## Figures and Tables

**Figure 1 molecules-27-05198-f001:**
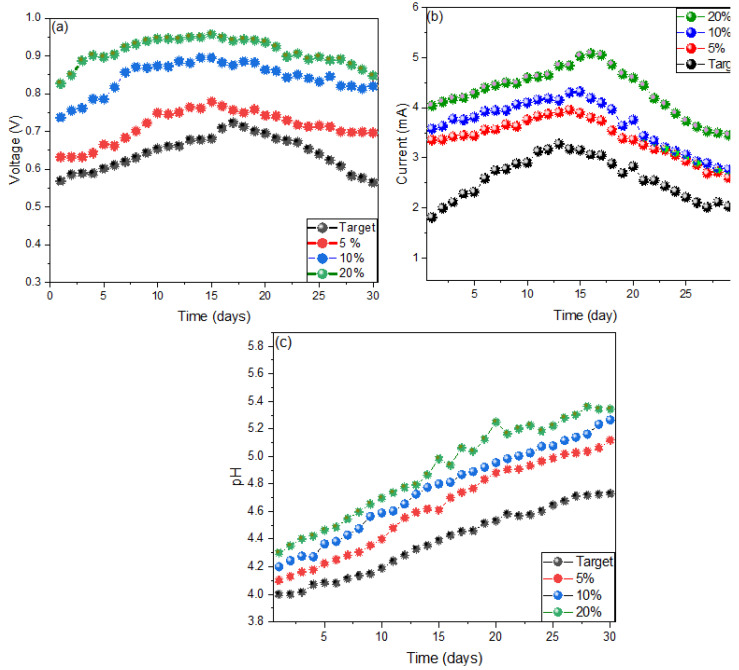
Values of (**a**) voltage, (**b**) electric current and (**c**) pH obtained from the monitoring of microbial fuel cells.

**Figure 2 molecules-27-05198-f002:**
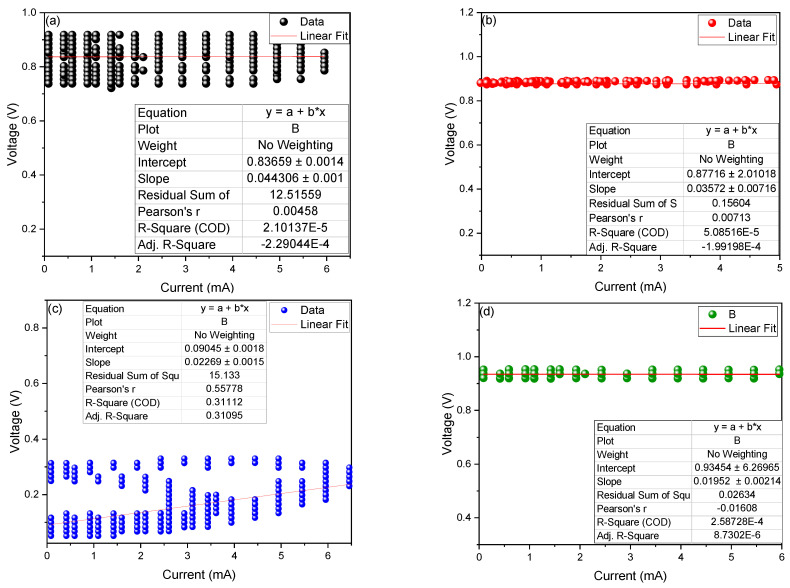
Internal resistance values of the microbial fuel cells at (**a**) 0, (**b**) 5, (**c**) 10 and (**d**) 20% sucrose.

**Figure 3 molecules-27-05198-f003:**
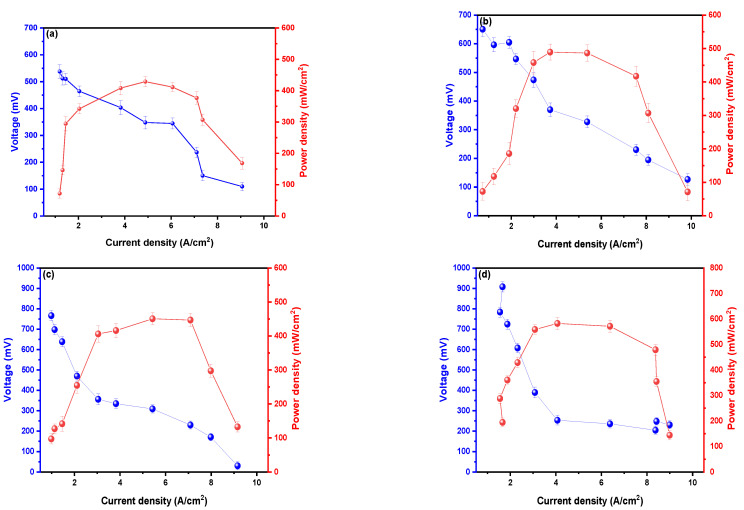
Values of the power densities as a function of the current density of the microbial fuel cells at (**a**) 0, (**b**) 5, (**c**) 10, and (**d**) 20% sucrose.

**Figure 4 molecules-27-05198-f004:**
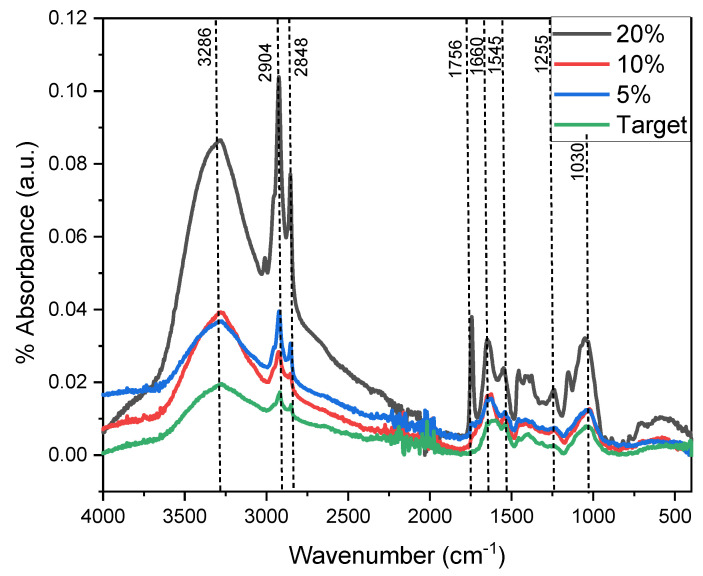
FTIR spectrophotometry of the papaya residues with saccharose.

**Figure 5 molecules-27-05198-f005:**
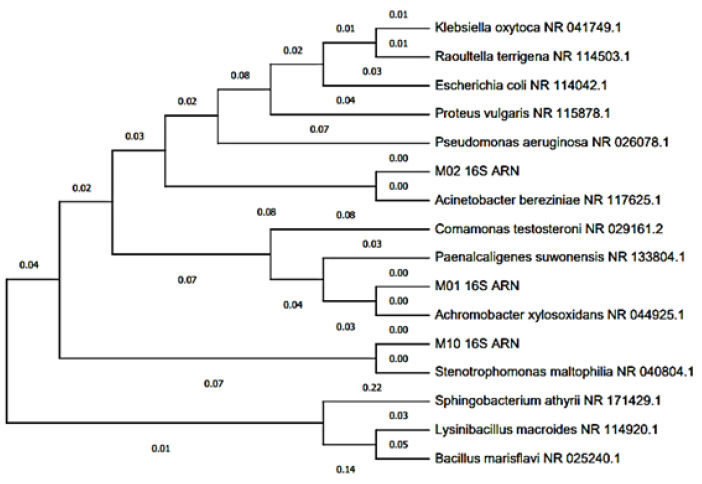
Dendrogram of bacterial clusters isolated from the MFCs anode plate.

**Figure 6 molecules-27-05198-f006:**
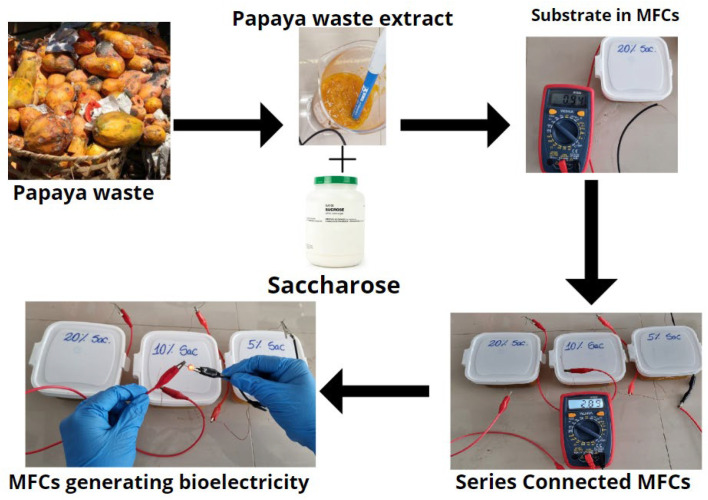
Electricity production in the MFCs.

**Figure 7 molecules-27-05198-f007:**
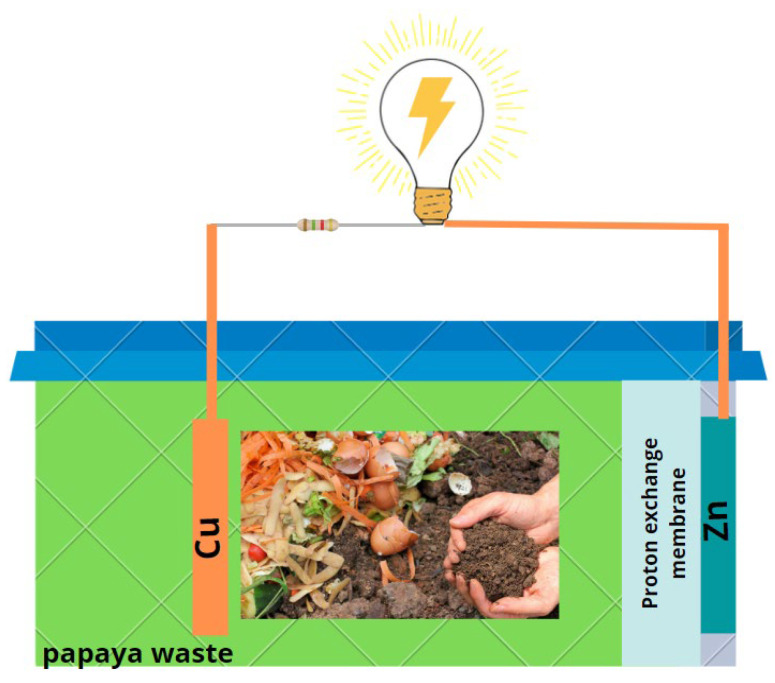
Schematization of single-chamber microbial fuel cells.

**Table 1 molecules-27-05198-t001:** BLAST characterization of the rDNA sequence of bacteria isolated from the MFCs anode plate.

BLAST Characterization	Consensus Sequence Length (nt)	% Maximum Identity	Accession Number	Phylogeny
** *Achromobacter xylosoxidans* **	1451	99.32%	CP053617.1	Cellular organisms; Bacterium; Proteobacteria; Betaproteobacteria; burkholderials; Alcaligenaceae; Achromo-bacter
** *Acinetobacter bereziniae* **	1468	99.93%	CP018259.1	Cellular organisms; Bacteria; Proteobacteria; Gammaproteobacteria; Pseudomonadales; Moraxellaceae; Acinetobacter
** *Stenotrophomonas maltophilia* **	1477	100.00%	NR_041577.1	Cellular organisms; Bacteria; Proteobacteria; Gammaproteobacteria; Xanthomonadales; Xanthomonadaceae; Stenotrophomonas; Stenotrophomonas maltophilia group

## Data Availability

Not applicable.

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
