# Peer review of "Electric Current Generation by Increasing Sucrose in Papaya Waste in Microbial Fuel Cells"

_molecules, 2022, doi:10.3390/molecules27165198_

Round 1
Reviewer 1 Report
please see the attachment.

Author Response
Dear colleague, thank you very much for the suggestions made.I send the answers of each suggestion:
- There are many format errors and typos, which made this manuscript to be not reliable. I recommend that the authors carefully examine all graphic and text formats and expressions throughout this manuscript. The errors and typos include but are not limited to the following:
- “peaks of 0.955 V. and 5,079 m…”, as highlighted, which is in line 20 and line 162, it is 5.079 mA according to Figure 2(b).
Ans. corrected
- In line 41. “population. and the environment…”, as highlighted.
Ans. corrected
- In line 65. “low cost Regarding the use of- materials,…”, loses a “.” before “Regarding”.
Ans. corrected
- In line 66. What is “MCCs”? Is it MFCs? If not, please give the full name of the MCCs.
Ans. corrected
- In line 117. “400 cm3” should be “400 cm3”
Ans. corrected
- In line 127. “Figure 01” should be “Figure 1”, which should be the same expression as others.
Ans. corrected
- In line 188. “Rint values. Found…”, as highlighted.
Ans. corrected
- In line 198. “(c) 10 and (c) 20% sucrose”, typos. The second “(c)” should be “(d)”.
Ans. corrected
- In line 204. “Mohamed et al. (2020) where I use kitchen”. Who is I?
Ans. corrected
- In line 209. “According to Yaqoob et al. 2020 These…”, the cite format is not correct and loses a “.” before “These”.
Ans. corrected
- The legends in all 3 graphs in Figure 2 do not coincide with the line or point color.
Ans. corrected
- The graphs in Figures 2-4 should be aligned. And I recommend that the authors can summarize the 4 graphs in Figure 2 in one graph since it would be better to analyze them when they are on the same scale.
Ans.The authors consider that placing it in the aforementioned way is the idea of using Ohm's law and the graphs would be seen very close together.
- From lines 121 to 122. How can the 10 mL of the proton exchange membrane be contained 400 mL H2O?
Ans. As a proton exchange membrane was used 10 ml of the solution obtained from 6 g of KCl and 14 g of agar in 400 mL of H2O .
- What will happen if more than 20% sucrose is added to the MFC?
- What will happen if more than 20% sucrose is added to the MFC?
Ans. This limit was used because at 30%, for example, an abrupt increase in the current and voltage values was no longer appreciated.
- What will happen if the 20% sucrose is added to the MFC without the papaya waste? Since sucrose is also an energy source for bacteria. The good results are due to the increase of the added sucrose, or the increase of the utilization efficiency of papaya should be deeply investigated.
Ans. Excellent contribution, in this investigation it was only considered due to the increase in sucrose. This will be considered for a future investigation.
- According to the cited articles, a higher concentration of phenols will generate a higher electrical current output. However, the FTIR results show that the most intense peak at 3289 cm-1(belonging to phenols) when using 10% sucrose is higher than those using 20% sucrose, which does not fit with the best performance when using 20% sucrose. The authors should deeply explain this phenomenon.
Ans. You are right, it was a mistake which has already been corrected.
- Several review papers were published recently regarding the valorization of waste. I urge the author review them and include any relevant discussion into the manuscript. For example: Emerging waste valorisation techniques to moderate the hazardous impacts, and their path towards sustainability, by Wang.
Ans. Done, reference [48]
best regards.

Reviewer 2 Report
I reviewed the article with the title ``Electric current generation by increasing sucrose in papaya waste``. The article topic is intriguing and promising in the area. Overall, the article structure and content are suitable for the Molecules journal. I am pleased to send you major-level comments, there are some serious flaws that need to be corrected before publication. Please consider these suggestions as listed below.
1. The title is not suitable, please must write the microbial fuel cells in the title.
2. Abstract need revision to add about MFCs. Please add one more introductory line of your objective at beginning of the abstract.
3. Keywords seems to be fine.
4. Introduction section must be written on more quality way, i.e., more up-to-date references addressed. The introduction needs major revision thoroughly. Page 1 Line 33 to 61, this paragraph is not directly related to the main objective, please concise it into few lines only. Please clear your focus.
5. The novelty of the work must be clearly addressed and discussed, compare previous research with existing research findings, and highlight novelty.
6. Research gap should be delivered in a clearer way with the directed necessity for future research work.
7. Why author choose papaya waste? Although Papayas contain high levels of antioxidants.
8. Is it MFC or MFCs???? Please check the abbreviations of words throughout the article. All should be consistent. Please revise your paper accordingly since some issue occurs on several spots in the paper.
9. The main objective of the work must be written in the clearer and more concise way at the end of the introduction section.
10. Overall, introduction need major level revision. Unnecessary talk is there with no meaning. Please stick to your objective. What is actual problem statement of your study?
11. English need major level revision, Page 2 Line 62 to 67, is single sentence. Please reduce the length of your sentences and maintain the actual meaning. Please revise your paper accordingly since some issue occurs on several spots in the paper.
12. Page 2 Line 62. At the end of this sentence (and microbial electrosynthesis cells among other bioelectrochemical systems) add a full stop with this reference. A glimpse into the microbial fuel cells for wastewater treatment with energy generation.
13. Please provide space between numbers and units. Please revise your paper accordingly since some issue occurs in several spots in the paper.
14. Please write the units in correct way such as cm3?
15. Why there is a fluctuation in voltage generation trend? State a reason.
16. Is it batch mode or continues system?
17. What about the external resistance? Is it fixed through the operation? How author select?
18. Why author did not consider basic electrochemical characterization such as EIS, CV. How you can justify your results without these. If you cannot perform, then add a comparative profile section to improve the article standard.
19. Regarding the replications, the authors confirmed that replications of the experiment were carried out. However, these results are not shown in the manuscript, how many replicates were carried out by experiment? Please, clarify whether the results of this document are from a single experiment or from an average resulting from replications. If replicated were carried out, the use of average data is required as well as the standard deviation in the results and figures shown throughout the manuscript. In the case of showing only one replicate explain why only one is shown and include the standard deviations.
20. How many bacterial species founds at anode? What about cathode?
21. Table and other formatting should be consistent. Please check.
22. Please add a comparative profile section to compare your results. Please consider these articles to cite in introduction and in comparative discussion to meet the journal standard. (i) Electricity generation and heavy metal remediation by utilizing yam (Dioscorea alata) waste in benthic microbial fuel cells (BMFCs) (ii) Application of rotten rice as a substrate for bacterial species to generate energy and the removal of toxic metals from wastewater through microbial fuel cells (iii) Local fruit wastes driven benthic microbial fuel cell: A sustainable approach to toxic metal removal and bioelectricity generation (iv) Potato waste as an effective source of electron generation and bioremediation of pollutant through benthic microbial fuel cell (v) Oxidation of food waste as an organic substrate in a single chamber microbial fuel cell to remove the pollutant with energy generation.
23. Where is section 4? Please check the formatting. Its also have serious errors.
24. Section 5 should be renamed by Conclusion and Future perspectives. The conclusion section is missing some perspective related to the future research work, quantifying the main research findings, and highlighting the relevance of the work with respect to the field aspect.
25. To avoid grammar and linguistic mistakes, Extensive level English language should be thoroughly checked. Please revise your paper accordingly since several language issues occur in several spots in the paper.
26. Reference formatting needs careful revision. All must be consistent in one formate. Please follow the journal guidelines.
Author Response
Dear colleague, I hope you are in good health; I send the corrections made to each of your suggestions:
- The title is not suitable, please must write the microbial fuel cells in the title.
Ans. Electric current generation by increasing sucrose in papaya waste in microbial fuel cells
- Abstract need revision to add about MFCs. Please add one more introductory line of your objective at beginning of the abstract.
Ans. Done
- Keywords seems to be fine.
Ans. ok
- Introduction section must be written on more quality way, i.e., more up-to-date references addressed. The introduction needs major revision thoroughly. Page 1 Line 33 to 61, this paragraph is not directly related to the main objective, please concise it into few lines only. Please clear your focus.
Ans. rewritten intro
- The novelty of the work must be clearly addressed and discussed, compare previous research with existing research findings, and highlight novelty.
Ans. rewritten intro
- Research gap should be delivered in a clearer way with the directed necessity for future research work.
Ans. In the conclusions part, a paragraph was placed for future work and the introduction was improved.
- Why author choose papaya waste? Although Papayas contain high levels of antioxidants.
Ans. Basically because we had done a previous investigation with good results, and we wanted to increase the electrical values. This is also due to the fact that in our country the export of this fruit has increased exponentially, generating a large amount of waste.
- Is it MFC or MFCs???? Please check the abbreviations of words throughout the article. All should be consistent. Please revise your paper accordingly since some issue occurs on several spots in the paper.
Ans. corrected
- The main objective of the work must be written in the clearer and more concise way at the end of the introduction section.
Ans. rewritten intro
- Overall, introduction need major level revision. Unnecessary talk is there with no meaning. Please stick to your objective. What is actual problem statement of your study?
Ans. rewritten intro
- English need major level revision, Page 2 Line 62 to 67, is single sentence. Please reduce the length of your sentences and maintain the actual meaning. Please revise your paper accordingly since some issue occurs on several spots in the paper.
Ans. rewritten intro
- Page 2 Line 62. At the end of this sentence (and microbial electrosynthesis cells among other bioelectrochemical systems) add a full stop with this reference. A glimpse into the microbial fuel cells for wastewater treatment with energy generation.
Ans. ok
- Please provide space between numbers and units. Please revise your paper accordingly since some issue occurs in several spots in the paper.
Ans. ok
- Please write the units in correct way such as cm3?
Ans. ok
- Why there is a fluctuation in voltage generation trend? State a reason.
Ans. When using copper as a cable for the measurement and being exposed to the environment, these cables tended to oxidize and each time for the measurements it had to be tied so that it is totally clean for the measurements, in other works considering platinum as a conductor cable, the measurements are but you counted, if a lot of fluctuation.
- Is it batch mode or continues system?
Ans. batch mode
- What about the external resistance? Is it fixed through the operation? How author select?
Ans. The external resistance was 100 ohms, the selection was due to the literature review is the most convenient.
- Why author did not consider basic electrochemical characterization such as EIS, CV. How you can justify your results without these. If you cannot perform, then add a comparative profile section to improve the article standard.
Ans. Although the EIS, CSV characterization is important for the characterization of the substrates, the research was based on observing the influence of sucrose with the MFCs as a whole.
- Regarding the replications, the authors confirmed that replications of the experiment were carried out. However, these results are not shown in the manuscript, how many replicates were carried out by experiment? Please, clarify whether the results of this document are from a single experiment or from an average resulting from replications. If replicated were carried out, the use of average data is required as well as the standard deviation in the results and figures shown throughout the manuscript. In the case of showing only one replicate explain why only one is shown and include the standard deviations.
Ans, Only one replica was used per percentage of sucrose, that is, they worked with 4 MFCs in total, one for each papaya waste with a percentage of sucrose. In future works it is recommended to carry out the work in triplicate but with a constant pH.
- How many bacterial species founds at anode? What about cathode?
Ans. In this research work we identify the species of the anodic electrode opted from the biofilms: Stenotrophomonas maltophilia, Acinetobacter iwoffi and Achromobacter xylosoxidans. Regarding the cathodic electrode, no identification was made because one side was exposed to the environment and the other to the agar used, and at the end of the monitoring the formation of any biofilms was not observed.
- Table and other formatting should be consistent. Please check.
Ans. Dear Colleague, it seems to me that table 1, if it is in the format.
- Please add a comparative profile section to compare your results. Please consider these articles to cite in introduction and in comparative discussion to meet the journal standard. (i)Electricity generation and heavy metal remediation by utilizing yam (Dioscorea alata) waste in benthic microbial fuel cells (BMFCs) (ii) Application of rotten rice as a substrate for bacterial species to generate energy and the removal of toxic metals from wastewater through microbial fuel cells (iii) Local fruit wastes driven benthic microbial fuel cell: A sustainable approach to toxic metal removal and bioelectricity generation (iv) Potato waste as an effective source of electron generation and bioremediation of pollutant through benthic microbial fuel cell (v) Oxidation of food waste as an organic substrate in a single chamber microbial fuel cell to remove the pollutant with energy generation.
Ans. For this investigation, only one cell per sample is considered, that is, for each percentage there is only one MFC.
- Where is section 4? Please check the formatting. Its also have serious errors.
Ans. corrected
- Section 5 should be renamed by Conclusion and Future perspectives. The conclusion section is missing some perspective related to the future research work, quantifying the main research findings, and highlighting the relevance of the work with respect to the field aspect.
Ans. Done
- To avoid grammar and linguistic mistakes, Extensive level English language should be thoroughly checked. Please revise your paper accordingly since several language issues occur in several spots in the paper.
Ans. Done
- Reference formatting needs careful revision. All must be consistent in one formate. Please follow the journal guidelines.
Ans. Done
best regards

Round 2
Reviewer 1 Report
The author has revised the paper sufficiently with my suggestion. I have no other comments.
Reviewer 2 Report
Accepted
This manuscript is a resubmission of an earlier submission. The following is a list of the peer review reports and author responses from that submission.
Round 1
Reviewer 1 Report
Dear authors, I'm afraid I will have to reject your paper for the following reasons:
The authors should do an in-depth literature review concerning applying this plant microbial fuel cell in various fields in the introduction section.
Zn and Cu are not appropriate as electrode materials for MFC because of their toxicity, especially that of Cu. From basic electrochemistry, it is well-known that in any galvanic cell Zn serves as an anode and Cu as a cathode due to their place in the activity series of metals (E0(Zn2+/Zn) = -0.76 V; E0(Cu2+/Cu) = +0.34 V ).From both points of view, it is absolutely inadequate to place the substrate at the compartment with the Cu electrode, which in the couple with Zn serves as a cathode!
Important routine analyses are omitted in the present study such as coulombic efficiency tests, cyclic voltamogramms, and the evolution of chemical oxygen demand throughout time.
Please address the above and please submit in the future.
Reviewer 2 Report
1. The results need more support with references and previous studies
2. The experiment was not explained and explained in an interpreted way for the ease of the researchers
3. The results need to standardize the variables and their colors as in the attached file
4. The experience requires a clear engineering and schematic drawing

Reviewer 3 Report
Main question addressed by the research: The work addresses the influence of sucrose in the generation of bioelectricity in papaya waste.
Originality and relevance of the topic: The topic is relevant to the field and it considers a suitable research gap.
Added value of the paper: The manuscript takes into account the study of different concentrations, and pH, however the main purpose of it is not clearly stated. The paper should include what aspects are critical for these assessments and clearly explain why they are analysing those and why they are needed at the end of the Introduction.
Quality of figures: Formatting should be consistent.
Specific improvements for the paper to be considered:
- Materials and methods contains a brief description of the microbial fuel cell and this information should be expanded. Operational range?
- Is there an interaction between the parameters studied?
- Discussion is weak and there is no connection with much literature.
- Would it be worth analysing the results instead of each individually? Density, voltage and current in the same figure?
- The conclusions are poor and they would need more elaboration so they clearly match the results.